# Accounting for multiple forcing factors and product substitution enforces the cooling effect of boreal forests

3

4 Eero Nikinmaa<sup>1†</sup>, Tuomo Kalliokoski<sup>1,2</sup>\*, Kari Minkkinen<sup>1</sup>, Jaana Bäck<sup>1</sup>, Michael Boy<sup>2</sup>, Yao

5 Gao<sup>3</sup>, Nina Janasik- Honkela<sup>4</sup>, Janne I. Hukkinen<sup>4</sup>, Maarit Kallio<sup>5</sup>, Markku Kulmala<sup>2</sup>, Nea

6 Kuusinen<sup>1</sup>, Annikki Mäkelä<sup>1</sup>, Brent D. Matthies<sup>1</sup>, Mikko Peltoniemi<sup>5</sup>, Risto Sievänen<sup>5</sup>, Ditte

7 Taipale<sup>1, 2,7</sup>, Lauri Valsta<sup>1</sup>, Anni Vanhatalo<sup>1</sup>, Martin Welp<sup>6</sup>, Luxi Zhou<sup>2</sup>, Putian Zhou<sup>2</sup>, Frank

8 Berninger<sup>1</sup>

9 <sup>1</sup> Department of Forest Sciences, University of Helsinki, P.O. Box 27, FI-00014 University of Helsinki, FINLAND.

10 <sup>2</sup> Department of Physics, University of Helsinki, P.O. Box 64, FI-00014 University of Helsinki, FINLAND.

<sup>3</sup> Finnish Meteorological Institute, P.O. Box 503, FI-00101 Helsinki, FINLAND.

<sup>4</sup> Department of Social Research, University of Helsinki, P.O.Box 16, FI-00014 University of Helsinki, FINLAND.

<sup>5</sup> Natural Resources Institute Finland, Viikinkaari 4, FI-00790 Helsinki, FINLAND.

<sup>6</sup> FB Wald und Umwelt Hochschule für nachhaltige Entwicklung Eberswalde, Schicklerstraße 5, 16225 Eberswalde,

15 GERMANY.

16 <sup>7</sup>Estonian University of Life Sciences, Department of Plant Physiology, Kreutzwaldi 1, EE-51014, ESTONIA

17 † Prof. Eero Nikinmaa was the initiator of this study. He regrettably deceased before we finished the paper.

18 \*Correspondence to: Tuomo Kalliokoski (tuomo.kalliokoski@helsinki.fi)

19

20 Abstract There is dispute over the climate change mitigating effect of boreal forest management due to the contrasting 21 influence it has on different vectors influencing radiative forcing (RF). For the first time, this study has combined the 22 estimated effects of carbon sequestration in forests and wood products, the surface albedo of forests, the direct and 23 indirect forcing of secondary organic aerosols and the avoidance of fossil emissions by product substitution, both in 24 the current and predicted 2050 climate. The aerosol effect was comparable in magnitude to that of carbon sequestration 25 and increased in importance in a warmer 2050 climate. Harvesting decreased the formation of climate cooling aerosols. 26 The aerosol effect was also larger than the opposing impact of increased surface albedo due to clear cutting in conifer 27 forests. When all above mentioned RF factors were accounted for, the RF of conifer-dominated stands was less 28 negative than that of broadleaf-dominated stands, despite the higher carbon sequestration of the former. Considering 29 also the cooling effect of product substitution, the differences in the RF impact of management alternatives that 30 maintained or increased forest biomass were small. However, the outcome depended heavily on the wood use pattern 31 and the assumed product substitution. A substantial increase in harvest with a clear increase in the share of small 32 dimension fiber and fuel wood use led to a clear climate warming effect in the simulations.

#### 34 1 Introduction

The boreal biome comprises 1/5<sup>th</sup> of the terrestrial carbon sink and about a third of terrestrial carbon stocks (Pan et al. 35 2011), and its associated effects on surface albedo (A) and secondary organic aerosols (SOA) may have a considerable 36 37 climate impact (Betts 2000, Tunved et al. 2006). Recently, the cooling impact from afforestation in the boreal zone 38 has been questioned, because models have indicated that the climate warming effect due to increased radiative forcing 39 (RF) from boreal forests' low albedo (Betts 2000) may exceed their cooling impact by carbon sequestration (Avila et 40 al. 2012). However, this argumentation does not account for the possible climate cooling effect of boreal forests via 41 their influence on SOA formation, nor product substitution. Forest ecosystems are producing large amounts of 42 Biogenic Volatile Organic Compounds (BVOC), which leads to SOA production and a greater climatic cooling effect 43 (Tunved et al. 2006) than, e.g., grasslands. Different forest management strategies (e.g. variations in harvest intensity, 44 selection of regenerated species and length of rotation period) can have significant impacts on carbon sink and storage 45 (Pihlainen et al. 2014), as well as on the non-carbon effects, i.e., albedo (Matthies and Valsta 2016) and SOA 46 formation.

47

Over half of the boreal forest area is currently managed or commercially operable, providing up to 17% of the global 49 industrial roundwood harvest (Burton et al. 2010, Gauthier et al. 2015). Since wood products can be used for 50 substituting more carbon intensive non-wood alternatives, the use of forests provides a potentially very useful and 51 cost-efficient tool for mitigation of greenhouse gas emissions (Sathre and O'Connor 2010). The current harvest level 52 of European forests is substantially lower than their wood increment (Forest Europe...2011), which could be seen as 53 an opportunity to increase biomass utilization to replace fossil fuels. Given the multitude of climate impacts associated 54 with forest management, an assessment of its net influence cannot, therefore, only be reduced to carbon storage or the 55 substitution of fossil fuels by bioenergy. A more comprehensive consideration is required that includes both 56 substitution and non-carbon effects.

In this study, we evaluate four key climate impacts of forest management: (1) carbon sequestration (in trees and soil, 59 i.e. forest ecosystems, and harvested wood products), (2) surface albedo of forest area (A), (3) forest originating 60 aerosols (SOA), and (4) avoided  $CO_2$ -emissions from wood energy and product substitution (PS). We calculate the 61 net of these effects at both a single stand and regional level, and also consider the effect on global climate if this 62 management is scaled up to the boreal forest level (Fig. 1). Our aim was to have a more comprehensive analysis of 63 climate impacts than the carbon balance. We were particularly interested in on the climatic balance of forest 64 originating SOAs and forest related albedo, and how the avoided CO2 emissions through product substitution 65 compensate the carbon sink reduction caused by wood harvesting. Finland was used as a case study region based on the wide availability of data and validated models for predicting the dynamics of managed boreal forests (Salminen et 66 67 al. 2005, Mäkelä et al. 2008a,b, Tuomi et al. 2011, The Finnish Statistical...2014).

#### 71 2 Materials and Methods

#### 72 2.1 Overall description of the study

First, we simulated RF for boreal forest management in Finland at the stand level for three dominant boreal tree species 74 in single species stands (Table 1, Scots pine (Pinus sylvestris), Norway spruce (Picea abies), and silver birch (Betula 75 pendula)), and for different site fertility over 100 years under both the current and projected 2050 climate (SRES A2 76 scenario, ensemble median climate model inmcm3.0, IPCC 2007) (Fig. 1, a and b). Then we estimated RF 77 development at the regional level in Finland, initializing the simulations with recent forest inventory data (The Finnish 78 Statistical... 2014) for both climate conditions (Fig. 1, c). Forest management scenarios were expressed as the ratio 79 of forest harvest to the present current annual stem wood increment (CAI, same absolute harvest levels also in 2050 80 climate) at the regional level (Fig. 1, d). For stand level simulations, the local impact of a management change was 81 expressed as the change in RF at the top of atmosphere per square meter of forest. The RF values were then compared 82 to a forest just after clear-cutting (bare land) which was used as a reference here (year 0, Fig. 2 and Fig. 4, a and b). 83 We also calculated cumulative radiative forcing over time (CRF) for illustrating the cumulative effect of separate 84 forcing agents (Fig. 4, c and d). For regional level simulations we reported the change in global RF due to the changes 85 of Finnish forests from their current state (Fig. 5, The Finnish Statistical... 2014).

### 87 2.2 Estimation of the forest development and the radiative forcing effects of carbon sequestration

Forest growth in Finland under the current climate was simulated with MOTTI, which is an empirical stand-level 89 analysis tool and decision support system for forest management (Salminen et al. 2005). MOTTI was used for 90 calculating tree growth dynamics for pure stands of Norway spruce, Scots pine and silver birch for site fertility types 91 in 3 classes (fertile, medium fertile). For each combination of tree species and site fertility, the forest 92 management practices spanning a stand's whole rotation in the simulations followed the recommendations for private 93 forest owners in Finland (Table 1, Recommendations for...2006). These recommendations are implemented in 94 MOTTI system and could be selected as an option in simulations. In the simulations, each stand type was artificially 95 regenerated by planting with site specific stand density. Thereafter, the tending of saplings, pre-commercial thinning, 96 commercial thinnings and final harvest was performed according to the recommendations. The timing of the pre-97 commercial thinning was based on the site specific tree height while the timing of the commercial thinning depended 98 on the site specific basal area limit defined in the recommendations. Whenever this limit was crossed a thinning from 99 below was performed. In the first commercial thinning, the opening of strip roads was mimicked by removing 18 % 100 of the stand basal area. The treatment intensity was defined on the basis of the basal area in such a way that removal 101 of trees decreased the stand basal area to the recommended level. The timing of final harvest was based on either stand 102 age or stand mean diameter (Fig. 2, Table 1). Final harvest was done in a conventional Finnish way i.e. as a clear cut 103 in which trees are topped, limbed and stems cut to the pre-defined lengths at the harvesting site. Harvesting residues, 104 i.e. tree tops, removed branches and stumps were left at the site. The simulation setup covered about 94% of the 105 upland forest growing sites in Finland (The Finnish Statistical...2014). Litter input into soil was calculated by 106 multiplying simulated biomass compartments and turnover rates in MOTTI with biomass expansion factors (BEF) 107 used in the Finnish greenhouse gas inventory (Official Statistics of Finland 2017). The soil decomposition model

(1)

YASSO07 (Tuomi et al. 2011) was used to simulate soil carbon dynamics with the obtained litter inputs. The carbon 109 stores and decay of harvested wood products were computed from the timber harvest assortment information that MOTTI simulates, and species-specific product distributions and life cycle information. For sawlogs, the allocation 110 of roundwood to industrial products was based on Karjalainen et al. (1994) because that source enabled species-wise 111 computations. Karjalainen et al. (1994) was the most detailed source available and resulted in the breakdown of 112 113 production given in Table 2. Pulpwood-sized wood was added to fibre products from sawlog residues and consumed for paper and packaging products. Storage of carbon in products was based on four lifecycle categories with 114 115 exponential decay, given in Karjalainen et al. (1994). 116

The total uncertainty of modeled forest carbon dynamics was assumed to be 15% in our analysis. This value consists
of parametric uncertainty of MOTTI (Salminen et al. 2005) and Yasso (Tuomi et al. 2011) models, and errors in BEFs
between different stand age classes.

121The marginal annual change of RF due to annual carbon stock changes (in trees, soil and harvested wood products)122was estimated following Lohila et al. (2010). The RF is defined here as a local change in the radiative energy balance123(W m<sup>-2</sup>) at the tropopause per one square meter in response to the change of the state of 1 m<sup>2</sup> of forest area for the124stand-wise simulations. The RF estimates due to the annual changes in biomass were converted to CO<sub>2</sub> equivalent125emissions/sinks assuming 50% carbon concentration in dry woody biomass. The decay of atmospheric CO<sub>2</sub> emissions126was estimated using the lifetime function *f*, (IPPC 2007):

$$f(t) = a_0 + \sum_{j=1}^3 a_j e^{-t/\tau_j}$$

where *t* is time,  $a_{j,j} = 1$ , 2, 3 are weights, and  $\tau_j$  are time constants (IPCC 2013). From these the cumulative changes 131 in atmospheric CO<sub>2</sub> due to different harvest scenarios were estimated and dynamics of RF impact were derived. 132

To illustrate the cumulative warming impact of forest management within a defined timeframe (*T*) we calculated the cumulative radiative forcing (CRF) ( $F^{C}$ ) as:

$$F_i^C(T) = \int_0^T F_i(t) dt$$
 (2)

Here  $F_i$  is RF related to effect *i* (CO<sub>2</sub>, A, SOA and PS) and T = 100 yr. Accordingly, the total CRF is: 139

$$F_{tot}^{\mathcal{C}}(T) = \sum F_i^{\mathcal{C}}(T)$$
(3)

For the landscape level simulations we estimated the impact of the management of the whole forestry area of Finlandfor the whole globe (Sect. 2.7).

#### 144

#### 145 2.3 Estimation of the climate change effects on the forest carbon sequestration

To analyse forest growth and carbon sequestration under climate change, the process-based models OptiPipe (Mäkelä 146 147 et al. 2008b) and PRELES (Mäkelä et al. 2008a) were used. In the OptiPipe model, tree volume growth was obtained 148 through the allocation of gross primary production (GPP) depending on the C:N ratio, temperature sum and soil 149 nitrogen availability. Soil nitrogen availability increases concurrently with the increasing temperature sum. The 150 potential increase of soil nitrogen availability was set higher in simulations for fertile forest sites than for poorer sites 151 (Mäkelä et al. 2008b). The GPP prediction in PRELES is based on the concept of light use efficiency. The model uses 152 daily values for the absorbed photosynthetically active radiation (aPAR) and modifiers for atmospheric CO<sub>2</sub> 153 concentration, light, temperature, vapor pressure, soil water and phenological state for GPP calculation (Peltoniemi et 154 al. 2015). The obtained relative changes in volume growth were used to modify the growth functions of MOTTI for 155 the 2050 climate.

The GPP potential of Finnish forests, used as an input in OptiPipe, was estimated using PRELES for the mean climate 158 model projections of SRES A2 climate scenario (IPCC 2007) (between RCP6.0 and RCP8.5 scenarios, Rogelj et al. 159 2012) for the year 2050. The constructed climate scenarios combined a daily observed data set on a 10 x 10 km grid 160 over Finland (Venäläinen et al. 2005) with changes in the long-term average, simulated by an ensemble of 8 Global 161 Circulation Models (GCM, CMIP3, Meehl et al. 2007) for the A2 emission scenario for 2011-2040, 2041-2070 and 162 2071-2100. The development of atmospheric CO<sub>2</sub> concentration during 21st century was obtained from the BERN 163 carbon cycle model (IPCC 2013). Details of the construction of climate scenarios are described by Rötter et al. (2013). 164 In the delta change method, the projected climate changes are added/multiplied (depends on the variable) to the 165 observed climate variables in the reference period. It is also assumed that the climate model bias remains the same in 166 the simulations of future climate (Meehl et al. 2007). Firstly, the monthly long-term changes in air temperature (T, 167 °C), precipitation (R, %), vapor pressure (hpa), global radiation (%) between the reference period (1971-2000) and future period for each GCM and emission scenario were calculated. The GCM grid cell center points were re-projected 168 169 to the projection of the grid of the observed database and monthly changes were bi-linearly interpolated to estimate 170 values for the center points of the 10 x 10 km grid cells. For each grid cell, monthly changes were linearly interpolated 171 to daily changes, which were added to the observed time-series.

## 173 2.4. Estimation of the radiative forcing effects of product substitution

The avoided CO<sub>2</sub> emissions related to wood utilization were computed separately for harvested sawlogs and pulpwood. For sawnwood, the substitution factors of individual studies listed in Sathre and O'Connor (2010) were used as data points to compute the average values for avoided carbon emissions per carbon in raw material. They were 0.913, 0.905, and 0.819, for Scots pine, Norway spruce and silver birch, respectively. For determining the uncertainty range of the values, standard errors of 0.566, 0.560, 0.507 were used for each species, respectively, calculated from the individual studies reported in Sathre and O'Connor (2010). For avoided emissions due to pulpwood use (Pingoud et al. 2010), a common value across species of 0.695 was used. Same CO<sub>2</sub> atmospheric life function (IPCC 2007) was

181 used as in case of forest carbon accounting (Sect. 2.2) for the decay of avoided emissions effect. Also RF due to the

- avoided emissions was estimated following Lohila et al. (2010).

#### 184 2.5 Estimation of the radiative forcing effects of surface albedo

Models of forest albedo were estimated for an area located in central Finland based on MODIS MCD43A3 blue-sky 186 albedos (Schaaf et al. 2002) and forest resource data produced by the Natural Resources Institute Finland (Tomppo et 187 al. 2008). Regression models were used to estimate the tree species specific forest albedos for different volume 188 thresholds utilizing information on the fractional covers of different forest types within the MODIS pixels (Kuusinen 189 2014). The land cover data were divided into five components: clear cut (growing stock  $\leq$  5 m<sup>3</sup>ha<sup>-1</sup>), young stand (pine 190 or spruce forest with growing stock > 5 m<sup>3</sup> ha<sup>-1</sup> but < 60 m<sup>3</sup> ha<sup>-1</sup>), pine forest (growing stock  $\ge$  60 m<sup>3</sup> ha<sup>-1</sup>), spruce 191 forest (growing stock  $\ge 60 \text{ m}^3 \text{ ha}^{-1}$ ) and deciduous broadleaved forest (growing stock  $> 5 \text{ m}^3 \text{ ha}^{-1}$ ). Species-specific 192 albedo for Scots pine and Norway spruce were assumed to follow a stepwise function during the total rotation. Albedo 193 was assumed to be highest in open clear cuts and linearly decrease for Scots pine and Norway spruce stands until the 194 growing stock reached 60 m<sup>3</sup> ha<sup>-1</sup>. Deciduous broadleaved species (mainly birches) albedo was noted to be insensitive 195 to changes in growing stock, so it was estimated as one value for the total rotation (stand volume  $> 5 \text{ m}^3\text{ha}^{-1}$ ). Monthly 196 means of component albedos were calculated for February-September, but due to low solar zenith angles during 197 winter, there were no good quality albedo retrievals available from October to January. For December and January, 198 the same albedo values were used as for February; October was given albedo equivalent to that of September and for 199 November albedo was linearly interpolated between the values of October (September) and December (February). 200 The resulting albedo values were translated into net shortwave radiation at the top of atmosphere using ECHAM5 201 radiative transfer model. The method is explained in more detail in Matthies et al. (2016). The uncertainty of the 202 albedo impact on RF was estimated to be 25%. This albedo uncertainty includes the differences between vegetation 203 types and in radiative transfer. The uncertainty of the RF from albedo of different forest types was estimated as a 204 proportion of their RF in a clear-cut area (used as a reference here). This assumes that changes in RF are directly 205 proportional to changes in albedo. The analysis is based on published values of RF and their errors (Table 1 in 206 Kuusinen et al. 2013).

207

## 208 2.6 Estimation of the radiative forcing effects of forest-originating aerosols

Measurements in a boreal Scots pine forest (SMEAR II, Hakola et al. 2012, Bäck et al. 2012, Aalto et al. 2014) and literature values of BVOC emission potentials for different species (Hakola et al. 1998, 2001, 2003, 2006, Smolander 210 211 et al. 2014) were used to simulate SOA formation for forests of different age and species composition using the one-212 dimensional chemical-transport model SOSAA (Boy et al. 2011). The model was constructed to reproduce boundary 213 layer transport, emissions, chemistry and aerosol dynamics. Since SOSAA is a column model, it assumes horizontal 214 homogeneity, with the consequence that simulations were carried out for individual tree stands under the assumption 215 that the gas- and particle phase compounds from one stand did not interact with other stands. The simulated aerosol 216 loading for the atmospheric boundary layer column was then employed to estimate the direct and indirect aerosol 217 induced RF. The indirect effect, which dominates the aerosol radiative effects, is based on the method by Kurtén et

al. (2003). The meteorological transport was based on the coupled plant-atmosphere boundary layer model SCADIS 219 (Sogachev et al. 2002, Sogachev and Panferoy 2006, Sogachev et al. 2012). The most important supporting equations are provided in Boy et al. (2011), and updates and validations are presented in Mogensen et al. (2015). The emissions 220 221 of organic vapors from the canopy were calculated by a modified version of MEGAN 2.04 (Guenther et al. 2006, 222 Smolander et al. 2014). The estimated emissions are highly dependent on meteorological factors (in particular 223 temperature and light), forest leaf area and leaf biomass, and furthermore, the emission potentials of individual organic 224 compounds that are specific for individual tree species (Mogensen et al. 2015). The data on Silver birch and Norway 225 spruce emission potentials was much less than those of Scots pine, and therefore simulations include more uncertainty 226 regarding these species.

The SOA effects were calculated for three different ages of pine, spruce and birch forest (Fig. 3 and Table 3) and the 229 values over the stand development were interpolated from those stages. The uncertainty range of SOA RF was adopted 230 from the study by Spracklen et al. (2008). They found a RF range from -1.6 to -6.7 Wm<sup>-2</sup> for the SOA effect for the 231 difference between no forest and closed canopy. Our simulations gave an average value for the same difference of -232 3.8 Wm<sup>-2</sup> when the current species distribution in Finland was assumed (50% Pine, 35% Spruce and 15% Birch). This 233 yielded an average range of  $\pm$  70% for the SOA effect. The aerosol dynamics module in the SOSAA model was based 234 on the University of Helsinki Multicomponent Aerosol model (UHMA, Korhonen et al. 2004) and described in a 235 recent publication by Zhou et al. (2014).

For the climate change effect in the SOSAA, the daily weather for year 2050 was predicted using the SRES A2 238 emission scenario and the csiro\_mk3\_5 climate model (0.34°C lower annual mean temperature in 2050 than projected by ensemble median model). The simulated meteorological parameters in SOSAA were then nudged towards these 239 240 predictions. The BVOC emission changes were calculated using the temperature dependency as expressed in MEGAN 241 2.04. Outputs from various climate models were used, in order to constrain SOSAA by the expected ambient 242 concentrations of O<sub>3</sub>, SO<sub>2</sub>, CO, NO, NO<sub>2</sub> and CH<sub>4</sub> in year 2050. The predicted mole fractions of SO<sub>2</sub> and O<sub>3</sub> were 243 obtained from GFDL-CM3 from the CMIP5 data archive (http://cmip-pcmdi.llnl.gov/cmip5/index.html). The 244 predicted mole fraction of CH<sub>4</sub> was taken from CESM1-WACCM (from NCAR). There exists no predicted mole 245 fractions of NO, NO<sub>2</sub> and CO for year 2050, therefore the annual mean emission rates of NO and CO from IPCC 246 RCP4.5 were utilized. It was assumed that there is an identical change in the NO<sub>2</sub> concentration as in the NO 247 concentration. This means that the 2010 measured concentrations were multiplied with the following factors: 248 O3•1.0472, SO2•0.3144, CO•0.3927, NO•0.2336, NO2•0.2336, and CH4•1.0562. The model was initialized with the 249 same background aerosol concentrations as in the current climate (year 2010) and constrained with the same 250 condensation sink as in 2010.

The changes in direct RF from aerosols were calculated between each canopy condition of the forest and species (Table 3). The method is based on the supplementary material of Paasonen et al. (2013) and on the article Lihavainen

et al. (2009).

## 255

The monthly mean cloud cover fraction was calculated from ECMWF reanalyzed low cloud cover fraction, which is also expected to have contributed towards the uncertainties. The changes in indirect RF from aerosols were calculated similarly as above between each canopy condition of the forest and tree species (Table 3) based on the method presented in Kurtén et al. (2003). The results in both direct and indirect forcing calculations were averaged for annual values in W m<sup>2</sup>, which represents the total amount of the radiative effect for 1 square meter of forest.

#### 262 2.7 Estimation of the radiative forcing effects of forest area simulations

The forest area simulations were calculated for the three main boreal species (Scots pine, Norway spruce and silver 264 birch, Sect. 2.2) with considerations for variations in site fertility and differences in age distribution in the forest. The forest data of the 11th Finnish national forest inventory (The Finnish Statistical...2014) were used to initialize the 265 266 simulations under the current climate, i.e. the current age structure of forests was used for the initialization. Four 267 different harvest scenarios were modeled, with harvest levels being relative to current annual increment (CAI) of the 268 forests (50%, 65%, 100% and 130% of CAI). The scenario of 65% harvest level corresponds the current harvest level 269 and was thus used as a reference for the other scenarios. The annually harvested volume consisted of volume from 270 recommended thinnings and from the final harvests. In order to obtain the targeted harvest level, the recommended 271 site-specific age of final harvest was delayed from the recommended minimum final felling age if needed. Shorter 272 rotation times than the recommended minimum final felling age were not used in the simulations. The resulting forest 273 area dynamics were coupled with stand level values (Sect. 2.2) of RF changes at different ages in order to obtain total 274 RF changes of forest area over the simulated time period, i.e. surface area of each species and site type combination 275 was multiplied with stand specific RF change curve (Fig. 4). The forest area RFs were areal integrals of stand RF 276 caused by carbon sequestration, surface albedo, aerosol and product substitution effects expressed relative to the initial 277 state of the forest. No spatial interaction between forest stands was assumed. Under the 2050 climate (Sect. 2.3), the 278 forests were initialized using the same site and age distributions as in the current climate. However, the initial volume 279 and growth of these classes corresponded to the new equilibrium in 2050 climate by modifying the growth functions 280 in MOTTI.

#### 282 3 Results

Forest management has a clear effect on radiative forcing (RF) regionally. The simulations show that both warming 284 and cooling can be attributed to management options, including the choice of tree species and the rotation period. The 285 RF was more negative with decreasing harvest levels, which is mainly attributable to higher carbon sequestration and 286 SOA formation. Substituting fossil fuels and fossil based material generated accumulation of avoided emissions which 287 reduced greatly differences between harvest schemes. At the stand level, the combination of all effects level resulted 288 in a large negative RF over the rotation (Fig. 4, Table 4). The 100-year average RF was increasingly negative with 289 increasing site fertility, and the contribution of SOA and PS together were over 70% of the total cooling impact in all 290 cases (Table 4). The inclusion of the SOA effect to RF assessment of the biosphere increased the negative RF of boreal

forests, i.e. climate cooling influence, and enforced the warming impact of forest harvesting. The cooling impact ofPS increased along the larger harvests but depended greatly on the substitution factors (Fig. 4).

In the SOA effect, the increased cloud albedo due to forest VOCs dominated since direct RF effect of aerosol particles ranged only from 3% to 6% in different species. The different effects had different dynamics with respect of forest development. Harvesting caused an increase in RF due to the loss of carbon sequestration and a decrease in SOA, but a decrease due to an increase in PS and also in A in the conifers. The deciduous silver birch stand's A had an opposite effect to that of the conifer stand's, being slightly higher than the open area A, i.e. clear cut had a warming effect in terms of A in silver birch. These differences between spruce and birch stands are illustrated more clearly by the cumulative RF (Fig. 4 c and d). The largest impact in CRF in all stands was caused by PS, due to its cumulative nature.

Under the 2050 climate, the SOA effect over stand rotation was larger than in the current climate (from -2.3 to -5.4
 Wm<sup>-2</sup> for the herb rich spruce site type) due to enhanced BVOC emissions and subsequent SOA formation, related to
 increased temperatures. This effect more than compensated for the increase in RF from reduced carbon sequestration
 (from -3.8 to -2.4 Wm<sup>-2</sup>) due to more rapid biomass turnover (litter and soil carbon, also shorter stand rotation time)
 in the warmer climate.

Forest management schemes at the regional level were simulated with different annual harvest intensities (50%, 65%, 309 100% and 130% harvest relative to the present CAI) using the modeled stand level increment curves. Totaling the 310 considered direct effects of forests (immediate A, SOA and CO<sub>2</sub>, excluding PS), meant that any harvest level below 311 100% of the current growth of Finnish forests, i.e., harvest schemes that increase the forest biomass from current, had 312 a cooling influence relative to the present state under the current climate (Fig. 5a). The difference between the lowest 313 and highest harvest levels of Finnish forests (50 vs. 130% of CAI) led to a net global RF difference of approximately 314 0.003 W over 50 years (the time period considered most important for climate mitigation actions) with the two lightest 315 harvest scenarios clearly cooling the climate and the most extreme harvest scenario clearly warming the climate from 316 the present (Fig. 5a).

When the product substitution (PS) effect was also considered, the differences in the RF impact between harvest levels 319 was reduced (Fig. 5b). Particularly, the RFs of the harvest intensities that maintained or increased the forest wood 320 stock were all within the assessed uncertainty ranges. The average substitution factors for saw logs and pulpwood (0.9 321 and 0.7 kg avoided C emissions per kg of C in wood, respectively) led to comparable RF influences over the first 50 322 years between PS and the other effects. The higher PS associated with a higher harvest rate largely compensated for 323 the lower direct cooling effect of SOA and  $CO_2$  so that the net difference between these harvest levels (50% - 100%) 324 was small (Fig. 5b).

The differences in RF between the harvest levels increased further under the 2050 climate. Largely due to changes inSOA forcing, the higher harvest rates had a bigger warming influence than in current climate, while change in the

328 difference between the lowest harvest rates was minor (Fig. 6). In the 2050 climate analyses, the PS was not considered 329 since it is a property of the technosphere (comprised of all of the structures that humans have constructed) and thus 330 larger and more rapid changes in it could occur than in the forest related properties of biosphere. Overall, the estimated 331 uncertainties for the non-carbon effects and PS exceeded those for forest carbon balances.

## 333 4 Discussion

Our results show that the rarely considered cooling via the formation of SOA as well as avoided emissions by product substitution (PS) play dominant roles in the climate forcing of boreal forests' management. Together, the climate 335 336 cooling impact of these two effects was about 70% of the average RF effect over stand rotation, but their response to 337 forest management was mutually opposite; harvesting increased cooling by the PS effect but decreased it by the SOA 338 effect. The analysis showed that a changing climate will lead to an increased importance of SOA. Aerosol and albedo effects were large enough to modify the carbon sequestration based order of dominant species' climate change 339 340 mitigation potential, shifting the rankings of spruce and birch dominated stands. The CRF curves show explicitly how 341 fundamental the inclusion of SOA and PS effects is for the climatic impact of forest management decisions. The large 342 differences in regional RF values between different scenarios underline the importance of forest management as a 343 driver of climatic forcing.

Naudts et al. (2016) concluded that forest management favoring conifers has contributed to climate warming since 346 1750's, particularly in the European temperate forest. Their conclusion was based on carbon sequestration together 347 with albedo and evapotranspiration impacts. We show here that including the effects of forest management on SOA 348 formation acts in the same direction as Naudts et al. (2016) conclusion. The SOA-induced cooling of closed canopy 349 forest relative to the reference clearcut was comparable to that caused by carbon sequestration over stand rotation. 350 Although the maximum cooling by SOA over rotation was lower than that of cumulative carbon sequestration, the 351 average cooling was comparable since the SOA effect tracked the size of the canopy whereas the carbon effect 352 followed biomass accumulation in trees, litter and harvested wood products. For the coniferous species, the SOA 353 induced cooling exceeded the warming induced by low albedo of forest cover under the present climate. The SOA 354 effect thus increased the climate warming impact of forest harvesting even in the boreal region. The SOA-induced 355 cooling of birch exceeded that of conifers, enforcing the Naudts et al. (2016) conclusion, though the leaf biomass is 356 lower in silver birch than in coniferous species. These species-specific differences in RF are mainly caused by their 357 different emission rates of BVOCs (Mentel et al. 2009). The differences increased under the 2050 climate (the net 358 difference of local RF between birch and spruce in herb-rich sites increased from -2.2 to -10.6 Wm<sup>-2</sup> without the PS 359 effect) as the relative importance of carbon sequestration decreased due to more rapid carbon turnover, whereas the 360 BVOC emissions and SOA formation simultaneously gained importance. This suggests that tree species selection in 361 favor of broadleaf species with higher BVOC emission rates can be of equal or greater importance than species 362 selection for albedo proposed earlier (Betts 2000, Spracklen et al. 2008, Bright et al. 2014). This result bears significance for the vegetation climate feedback since growth predictions suggest that warming climate favors 363 364 broadleaf species over conifers (Kellomäki et al. 2008). We would like to point out that the uncertainty related to the

emission potentials of Norway spruce and especially silver birch is high due to very limited published data. 366 Unpublished preliminary results (completed after submission of this manuscript), however, suggest that the 367 monoterpene emission of silver birch used for this study could be in the order of 6-18 times too high. This would 368 decrease the aerosol RF of birch so significantly that conifers forests would be cooler than silver birch forest with 369 respect to aerosol RF.

These results strongly suggest that the SOA effect needs to be included in a full assessment of the climate impacts of forest management, however, the conclusions are conditional to our current understanding of SOA formation. Numerous atmospheric processes and feedback mechanisms influence the RF of aerosols. For example, it was recently discovered that the composition of aerosol size distribution is influenced by Highly Oxidized organic Molecules (HOMs) (Ehn et al. 2014, Jokinen et al. 2015). Further, the exact effect of clouds on climate depends on cloud altitude, which was not accounted for in our simulations. The average RF values due to SOA impacts are in the middle range of previously estimated values (Spracklen et al. 2008), which is reassuring and gives confidence to the estimates.

Our results suggest a weaker albedo impact than indicated previously (Betts 2000). However, the estimated albedo 380 effect was largest in relative terms in low fertility forests (Forest Resources...2000) which probably correspond better 381 to the Siberian forests where a large albedo impact has been estimated (Betts 2000). Nevertheless, our estimates of 382 the RF due to albedo changes were relatively small. Estimates of the albedo effects of deforestation of boreal forests 383 vary widely in the literature (Betts 2000, Bright et al. 2014, Alkama and Cescatti 2016). Similar to our study, 384 O'Halloran et al. (2012) also derived albedo values from MODIS data for stands of different ages but obtained a 385 somewhat larger albedo difference than our study between forest and no-forest patches. This was mainly due to lower 386 winter and spring albedo of mature forests than in the study applied here (Kuusinen et al. 2012). Our estimates are 387 averages of clear-cut areas and forest stands at different ages for large forest areas and should therefore be 388 representative of prevailing conditions in Finland. Nonetheless, the different studies may reflect differences in forest 389 structure and conditions and empirical data have uncertainties of its' own depending on the length of time series or 390 accuracy of inventory data etc. Therefore, caution must be used when interpreting the results for the whole boreal 391 forest. Overall, the importance of the springtime albedo effect will decrease in a warmer climate with a shorter snow 392 cover period.

The avoided emissions are pivotal for assessing the climate change mitigation efficiency of forest management. 395 Without it, the lower the harvest the larger the cooling climate impact. Conversely, numerous studies agree that in the 396 long run the avoided emissions of product substitution inevitably lead to climate cooling as they accumulate over time 397 (Sathre and O'Connor 2010). At the time of harvest, carbon stored in the forest is lost and, assuming a given demand 398 of products, fossil fuel emissions are reduced as wood products can act as substitutes. Our results show that in a region 399 of established forest management with sustained harvest without a reduction in the regional wood inventory (Gauthier 400 et al. 2015), a moderate decrease or increase of the harvest rate may not bring about large changes in net climate 401 benefits if the substitution effect is considered in the time range of 50 years. This result is highly sensitive to the