# Peer review of "Accounting for multiple forcing factors and product substitution enforces the cooling effect of boreal forests"

_Biogeosciences, 2017_

## Referee Comment (RC1) · Anonymous Referee #1 · 29 May 2017

In "Accounting for multiple forcing factors and product substitution enforces the cooling effect of boreal forests", the authors aim to assess the "climate change mitigating effect of boreal forest management" (L20, Abstract) of four different stand types in Finland by comparing the radiative forcing (RF) caused by $CO_2$ in forests and wood products, surface albedo, secondary organic aerosols (SOA), and product substitution for two scenarios: recommended forest management practices vs. a counterfactual bare land. The authors also contrast the results obtained under the current climate and a projected 2050 climate. The subject is timely and relevant. Unfortunately, as explained below, the study design is flawed, many methodological approaches are too crude to sufficiently trust the results, and the Methods lack clarity and explanations.

1. Study design. The main objective of the study is to estimate the RF due to boreal forest management (as noted in the first sentences of the Abstract and Discussion); consequently, the counterfactual scenario must be unmanaged forest for each of the four stand types studied, instead of a hypothetical bare land. Under the current approach, readers are misled into thinking that the results provided in Table 4 and Figures 4-6 correspond to the RF due to forest management, which is not the case. The Editor made a similar comment previously, to which the authors responded they were interested in comparing different stand types. Besides the misleading Table and Figures abovementioned, this response is inadequate because: 1) forest managers cannot realistically choose to transform one stand type into another (e.g. transform a mesic Norway spruce stand into a sub-xeric Scots pine stand); and 2) the indirect aerosol forcing is highly non-linear, which implies that the counterfactual scenario matters for the difference in SOA effect across stand types. Finally, even if the bare land counterfactual was appropriate for some reasons I am not grasping, it was not implemented properly in the study. First, the emissions of dust aerosols would be much higher for bare land than for forest and the resulting large RF could very well be larger than all the effects currently considered. Second, the GHG implications of the activities required to maintain the forest as bare land (periodic mechanical/chemical treatment, fate of the original forest carbon stocks, etc.) would also need to be considered.

2. Methodological approaches. First, the authors should have used a proper dynamic aerosol-climate model to compute the SOA RF. The approach used to compute the direct RF can only provide "[a]n order-of-magnitude estimate of the strength" (Paasonen et al., 2013) of the effect. Similarly, the approach of Kurten et al. (2003) used to compute the indirect RF is much too crude compared with state-of-the-art methods and only accounts for the 'cloud albedo' effect (this last shortcoming is not clearly mentioned). Moreover, the simulation setting (location(s) considered, input data, etc.) is unclear, but I doubt very much that these computations were site-specific across Finland – a major limitation given that "estimated emissions are highly dependent on meteorological factors (in particular temperature and light)" (L222-223), hence are highly site-specific.

Second, the approach for surface albedo also does not pass muster. Changes in albedo were estimated "for an area located in central Finland" (L185) instead of being estimated across the region studied; these changes were then "assumed to follow a stepwise function during the total rotation" (L192), as visible in Figure 4a,b (yellow curve), which is much too simplified compared with published results (e.g. Figures 1-2 of Amiro et al., 2006). Third, the results for the 2050 climate were computed for an unrealistic instantaneous change ("for the mean climate [..] for the year 2050"; L157-159) instead of for a realistic transient climate change.

3. Methods. The explanations provided in the Methods do not allow readers to understand how the study was performed and what the results really mean; here are only some examples. Unclear explanations: was the MOTTI stand-level forest model run across all stands (how many?) across Finland, only for the 12 combinations of stand type and site fertility, or for something else? Not enough details: how were the "0.913, 0.905, and 0.819, for Scots pine, Norway spruce, and silver birch, respectively" (L177) displacement factors obtained from the Sathre and O'Connor (2010) results exactly? Missing justifications: the 0.695 displacement factor for pulpwood based on Pingoud et al. (2010) seems to implicitly assume that 50% of the energy displaced is from coal and 50% from natural gas (their Table 2); why is such an assumption valid here? (In the first place, having displaced emissions from pulpwood should also have been explained; this rests upon a link Pingoud et al. (2010) simply assumed between pulpwood and bioenergy.) Elements that are not mentioned at all: what are the displacement factors for the other two categories appearing in Table 2 of the study, i.e. plywood and "process energy" (same as bioenergy, I assume)? Elements the authors apparently did not think about, but that should at least be acknowledged as limitations: not accounting for the methane emissions from products ending up in landfills or for the aerosols emitted from bioenergy.

References

Amiro et al. (2006). Agricultural and Forest Meteorology 140, 41-50
Kurten et al. (2003). Boreal Environment Research 8, 275-285

Paasonen et al. (2013). Nature Geoscience 6, 438-442

Pingoud et al. (2010). Silva Fennica 44, 155-175

Sathre and O'Connor (2010). Environmental Science & Policy 13, 104-114

—————————————————————

---

## Author Comment (AC1) · 21 Jun 2017

Reviewer comments with quotation marks, our answers without quotation marks.

1. Study design The reviewer statement that we compare forest management against bare land seems to be based on a misunderstanding. We present our analysis at two different spatial scales: a single stand and the regional scale: 1) For a single stand we start with "bare land" and look on how different forcings evolve over time. These forcings are compared to bare land since stand development at time 0 starts from bare land (this is the data used in Figure 2-4). The use of bare land for stand level simulations is justified since radiative forcing for bare land does not depend on tree

species. 2) Our reference, i.e. counterfactual, scenario, for the forest management scenarios (Figure 5 and 6) is the current level of harvesting which we compare to scenarios with either decreased or increased harvesting levels. Our opinion is, thus, that we can assess the climate impact of these forest management alternatives. We have rewritten the manuscript in several places in order to clarify this.

"...forest managers cannot realistically choose to transform one stand type into another (e.g. transform a mesic Norway spruce stand into a sub-xeric Scots pine stand);..."

Our analysis does not imply that kind of decision making. In our analysis, we distinguish between site types (we used three site types) and species. Site types roughly represent the potential of a site to produce wood under a given climate and depends largely on soil conditions. These site types are fixed in our analysis and cannot be changed. In intensive forestry, as in Scandinavia, forest managers can change species after a clearcut. Changes between Norway spruce and birch make sense on fertile sites while it makes no sense to grow these species on sub-xeric sites (the productivity would be very low). Therefore, we allowed changes in species compositions after clearcuts for the fertile sites (50% Norway spruce – 50% silver birch).

". . .the indirect aerosol forcing is highly non-linear, which implies that the counterfactual scenario matters for the difference in SOA effect across stand types."

We agree with the reviewer that our approach is rough. However, as we pointed out, our counterfactual scenario is not bare land i.e. Finland without forests. Our research was motivated by the need of forest managers which are typically made at the stand level (like do I cut now or later, do I plant spruce or birch). The proposed approach gives a first approximation of the aerosol effects of forest management for the stand scale. We acknowledge that the approach is potentially biased but approximately valid. Moreover, even more developed approaches would have a high uncertainty which is visible from the high uncertainty of aerosol forcing in the IPCC reports. For the landscape scale the importance of nonlinearities partially cancel each other out and get in any case diluted

since changes, e.g. in number of clear cut areas, are anyhow quite small.

2. Methodological approaches "…The approach used to compute the direct RF can only provide "[a]n order-of-magnitude estimate of the strength"…"

We agree that our methods for assessing aerosols are not as sophisticated as they could be and it's also true that they include a lot of uncertainty. A proper dynamic aerosol cloud model could have been used indeed. We are not, however, convinced that by using the models the reviewer suggests, the results would become much more trustworthy since the main uncertainties in the case of aerosols still stem from lacking the basic understanding of the aerosol formation processes and how they affect cloud formation. And we want to point out here that regional models more or less parameterize chemical and physical processes out of computational costs, which were not the case in our approach. Therefore, it is not possible to give precise estimates but we regard as reasonable to use an order-of-magnitude estimation in our case because the aim is to provide an assessment of a hypothetical analysis that incorporates many aspects, e.g. the rotation effects, all of which have large uncertainties. We want to remind that this is the first study where all these different climate agents are explicitly related to forest management. In future studies, our results could be compared against the results produced with other methods.

The reviewer also mentions that our aerosol results are site specific. This is partly true since we consider these results as species specific. By changing the tree species, the results may change drastically. Specific species grow on specific sites and here the reviewer is right. However, we used different site inputs (stand characteristics) in our VOC model in order to characterize the most common growing sites in Finland and thus we were able to generalize our results for the whole country.

"Changes in albedo were estimated "for an area located in central Finland" (L185) instead of being estimated across the region studied; these changes were then "assumed to follow a stepwise function during the total rotation"…"

It is true that our albedo analysis is based on for one area in Central Finland (covering about 1.4 million ha). Previous studies have shown low differences in summer albedo between northern and southern Finland (Kuusinen et al. 2012). The estimates for winter albedo for the northern area were considered to be not very representative because there were no good quality Modis images for our Northern test area. Our simplified step-wise function is based both on the results published in Kuusinen et al. (2013) and the albedo analysis done in this study showing that mean albedo values between regions within Finland are quite stable. Thus, we deduce that our approach produces stand level differences between species accurately enough to be able to generalize these for the whole country.

"...the results for the 2050 climate were computed for an unrealistic instantaneous change ("for the mean climate [..] for the year 2050"; L157-159) instead of for a realistic transient climate change."

We agree that transient climate change and a dynamic response of forest growth along that would be more realistic. The forest state at the beginning of the simulations in the 2050 climate, i.e. the volume of forest stands per hectare, would of course be different if the change would have been transient. However, the analysis of transient change was not possible due to the computational limits, mainly because of the complex VOC-aerosol modeling. In addition, due to intensive forestry, the main driver of change occurred in Finnish forests during the last century have not been climate but the decisions of forest managers concerning silviculture and harvests. Future projections corresponding of these kind of changes are very difficult to simulate in a transient way. Our aim was to illustrate the potential effect of climate on forest growth and how the climate impact of changing harvesting level in that new state, 2050 climate, could differ from the current one. We deemed that the adopted approach was suitable for this purpose.

3. Methods "The explanations provided in the Methods do not allow readers to understand how the study was performed and what the results really mean."

We have rewritten the methods part in order to clarify the sections that the reviewer points out.

". . .was the MOTTI stand-level forest model run across all stands (how many?) across Finland, only for the 12 combinations of stand type and site fertility, or for something else?"

Motti runs were performed for the combination of tree species and site types (Table 1 and Table 4) i.e. 12 descriptions of stands. These stand level results were then generalized for the whole country by using the proportions of these cases in Finnish forests based on National Forest Inventory data.

"..how were the "0.913, 0.905, and 0.819, for Scots pine, Norway spruce, and silver birch, respectively" (L177) displacement factors obtained from the Sathre and O'Connor (2010) results exactly?"

The factors are per cubic meter of sawlogs. The average sawnwood substitution factor for sawnwood based on Sathre and O'Connor was computed as an average of those studies that dealt with whole buildings or the construction sector. Data values that referred to individual products were discarded as they were considered to be case dependent. Accordingly, the omitted studies were Jönsson et al. 1997, Knight et al. 2005, Petersen & Solberg (2002), Petersen & Solberg (2003), Petersen & Solberg (2004), Sedjo (2002), Scharai-Rad and Welling (2002) window frames. Then, the arithmetic average substitution factor was 2.1. This was multiplied by the recovery rates of mechanical wood products (sawnwood and plywood) on Karjalainen et al. 1995: 0.435, 0.431, 0.39 for Scots pine, Norway spruce and birch, respectively.

"Missing justifications: the 0.695 displacement factor for pulpwood based on Pingoud et al. (2010) seems to implicitly assume that 50% of the energy displaced is from coal and 50% from natural gas (their Table 2)."

The average consumption of coal and natural gas in Finland during 2006 to 2015 has

been 152 000 and 127 000 TJ p.a. (www.stat.fi). We considered their amounts to be so similar that equal proportions would be a valid approximation, especially taking into account the future uncertainty.

"...(In the first place, having displaced emissions from pulpwood should also have been explained; this rests upon a link Pingoud et al. (2010) simply assumed between pulpwood and bioenergy.)"

The substitution computations all rely on the comparison of forest management alternatives that lead to comparisons of wood material produced. An associated assumption in these consequential life cycle assessments is that the societal need for materials/energy is fixed and the additional production is used for substituting fossil materials/energy. Another fossil alternative to pulpwood could also be plastic but this was not applied due to the lack of information about substitutability at large scale.

"Elements that are not mentioned at all: what are the displacement factors for the other two categories appearing in Table 2 of the study, i.e. plywood and "process energy" (same as bioenergy, I assume)?"

Plywood is combined with sawnwood in the amounts for mechanical wood products. Process energy was not accounted for as fossil emissions savings beacause it is included in the substitution factors, already.

"Elements the authors apparently did not think about, but that should at least be acknowledged as limitations: not accounting for the methane emissions from products ending up in landfills or for the aerosols emitted from bioenergy."

Yes, we agree with the reviewer that within one study everything is not possible to handle. We have added these limitations to the discussion.

---

## Referee Comment (RC2) · Anonymous Referee #2 · 22 Jul 2017

Review of Nikinmaa et al Biogeosciences

The goal of this paper is to make a more holistic evaluation of the net climate effects of Finnish boreal forests by adding secondary organic aerosols and product substitution to more traditional assessments including carbon and albedo, with radiative forcing as the metric. The study examines different management intensities and tree species under current and future climate scenarios. Overall, it is an interesting and extremely ambitious undertaking, and a logical and meritorious next step in the climate/land use literature. The authors have made good efforts to use the most regionally-specific methodologies. Unfortunately, the problem with such an ambitious analysis is that each

section essentially requires the level of detail of a full manuscript in order to be appropriately detailed. To condense the methodology, the authors heavily leverage prior work through frequent use of citations, and subsequently skimp on critical methodological details, making it impossible for the reviewer (or potential readers) to make accurate assessments of the methods. I think this is a fatal overall flaw of the manuscript, along with a few others I will review below.

Major issues

For the stand level assessment, the use of bare ground as a control is inappropriate. From the first sentence of the abstract, the goal of the paper is to examine the "climate change mitigating effect of boreal forest management". The appropriate control case is then whatever the natural land cover type would exist in the absence of management (e.g. climax unmanaged boreal forest).

As stated earlier, the inclusion of SOA is an obvious and important next step for this type of work, but unfortunately I simply do not believe the science is mature enough to be able to accurately represent reality in this sort of simplified column-wise treatment. If the uncertainty were properly handled and propagated through all the steps (which, critically, it is not here), then the large uncertainty would render the whole exercise moot. I'm extremely dubious about the methodology under a current climate, let alone under future climate with likely very different atmospheric chemistry. The assumptions that have to made are so numerous and crude that I think it renders the exercise meaningless. I think the only way to even attempt it is with a fully coupled GCM with MEGAN and online atmospheric chemistry etc., where many scenarios can be evaluated and error bounds generated. Furthermore, carbon and albedo can be more easily handed in the column-wise radiative forcing framework because we assume $CO_2$ is well mixed, and albedo RF essentially operates in 1D. But SOA formation (and evapotranspiration, for that matter) produce an RF in a very nonlinear and very 3D way in the atmospheric fluid flow in a complex interaction with clouds (in the indirect forcing case), and are not well mixed, and are thus not completely attributable to the forest processes themselves.

For these reasons I think SOA may remain a fundamentally difficult if not impossible problem to address using the 1D radiative forcing framework.

As I mentioned earlier, error and certainty is handled crudely. In several sections uncertainty ranges are mentioned, but it's not clear that these are propagated through the analysis.

It's not clear to me how the stand level and regional level analysis are different, necessary, and what we learn from doing both. The assumptions in the regional level analysis seem better, but I have serious concerns about the albedo data for that section.

Regarding albedo data, is it really possible to develop species specific albedo functions from MODIS data? Why is the result a step function? How do these values compare to those of Bright et al., (2014)? Impossible to properly evaluate without presenting these data as a figure.

The section starting on line 157 about how future climate scenarios is very confusing and impossible to follow.

Regarding the decision not to include product substitution in year 2050 because technology will change too much: to me I feel the same way about the SOA predictions for year 2050 because there is both uncertainty in the methodology and the future emission and atmospheric chemistry such that it is intractable.

Minor comments The abbreviation A for albedo is confusing since it is a commonly used letter. Perhaps alpha?

Reference Bright, R.M., Antón-Fernández, C., Astrup, R. and Strømman, A.H., 2014. Empirical models of albedo transitions in managed boreal forests: analysis of performance and transportability. Canadian Journal of Forest Research, 45(2), pp.195-206.